# Acrylamide-Formation Potential of Cereals: What Role Does the Agronomic Management System Play?

**Falko Stockmann** [1],*, **Ernst Albrecht Weber** [1], **Benjamin Mast** [1], **Pat Schreiter** [2], **Nikolaus Merkt** [1], **Wilhelm Claupein** [1] **and Simone Graeff-Hönninger** [1]

[1]  Institute of Crop Science, University of Hohenheim, D-70599 Stuttgart, Germany
[2]  Chemisches und Veterinäruntersuchungsamt Stuttgart, Schaflandstraße 3/2, D-70736 Fellbach, Germany
*   Correspondence: letsch.stockmann@gmail.com; Tel.: +49-9420-8010239

**Abstract:** As bakery products contribute considerably to the daily intake of the carcinogen acting substance acrylamide (AA), the aim of this study was to evaluate the impact of the management system (conventional vs. organic farming) on AA precursor levels of free asparagine (Asn) across different cultivars of the cereal species, namely winter wheat (*Triticum aestivum*), winter spelt (*Triticum aestivum* ssp. *spelta*) and winter rye (*Secale cereale*) with simultaneous consideration of gained grain yields and flour qualities. For this purpose, orthogonal field trials were established at two sites in Southwest Germany over two growing seasons (2006–2007 and 2007–2008). The results indicated a significant impact of the management system on free Asn contents in white flour. Across all species, free Asn contents in the white flour was 26% lower under organic compared to conventional farming. The impact of the management system on individual cultivars was obvious with a maximum reduction in free Asn contents of 50% in wheat cultivars if organically produced (e.g., for cultivars Ludwig, Privileg, Capo). For spelt, a significant impact of the management system was only found in 2008 with a reduction in free Asn of up to 25% if organically produced. Across both cropping systems, cultivar Franckenkorn reached the lowest levels of free Asn. For rye, a significant impact of the management system was observed only in 2007 with 33% higher Asn amounts in the conventional management system. Independent of the cropping system, rye reached the highest levels of free Asn followed by wheat and spelt. Depending on species, there was also an impact of the two systems on crude protein. The organically cropped wheat had a significantly lower level, but this was not observed for spelt and for rye only in 2007. The possible reason for the low free Asn content in the organically produced wheat flour could partially be the lower crude protein amount. Furthermore, the results indicated that lower AA contents in bakery products can be achieved by proper selection of species (e.g., 66% lower if rye is replaced by wheat) and cultivars. With an appropriate choice of the cultivar, a reduction of up to 65% was possible within wheat, along with a reduction of 44% within spelt and 12.5% within rye. In summary, the results indicated that organically produced wheat especially offers the opportunity to significantly lower the AA potential of bread and bread rolls by the choice of raw materials low in free Asn.

**Keywords:** acrylamide; free asparagine; management systems; organic; conventional; agriculture; cereals; species; cultivars; product quality

## 1. Introduction

Due to a current announced regulation of the European Commission [1] food industry and gastronomy face the challenge of establishing immediate mitigation strategies and benchmark levels for acrylamide (AA).

More than 15 years after the first findings of the food-borne toxicant AA in starch-rich heated foods, the dietary intake of AA is now seriously regarded to potentially increase cancer risk for humans [2]. Therefore, AA in foodstuffs should be minimized to a level as low as reasonably achievable.

Since 2002, it was successfully shown that AA is formed during a thermal treatment of carbohydrate rich food like cereals and potatoes [3,4], where 'reducing sugars' (mostly glucose and fructose) react with the amino acid free asparagine (Asn) within the Maillard reaction [5]. While reducing sugars are the limiting precursor in heated potato products [6], free Asn is the limiting precursor for processed cereal based products [7–10]. Although strongly heated potato products can contain much more AA than cereal based products, foods like bread, rolls, biscuits and crisp bread contribute to about 25% to 45% of the dietary AA intake in Germany [11]. This is mainly due to the high daily per capita consumption of bread of almost 240 g [12].

It has been shown that the amount of AA can be minimized by adjusting processing parameters such as pH, heating temperature and time of heating, changing baking agents, as well as adding additives, or by elucidating the mechanistic pathways of AA formation and eliminating precursors or intermediates [6,12–17].

Modification of recipes or processing conditions often led to significant reductions in AA levels of the final products. However, their applicability is restricted because their use is often accompanied by negative effects on taste or quality of the final products, or their use is simply too expensive. The used raw material, notably the Asn precursor content in cereal grains, plays an important role for the AA formation potential. Consequently, one alternative to reduce AA in the final product might be minimizing Asn contents in cereal grains by agronomic management.

Several studies showed that cereal species differ in their Asn levels and in consequence in their acrylamide formation potential. Rye usually has higher Asn levels compared to wheat and spelt [7,8,18]. Moreover, cultivars can differ considerably in their precursor content as shown by several studies [7,8,18–20]. Taeymans et al. [20] reported a fivefold range for a variety of European wheat cultivars, and Claus et al. [7] found a variability of Asn contents in nine German winter wheat cultivars of up to a factor of 3. Corol et al. [19] reported differences of 150 wheat genotypes analysed as wholemeal samples of almost 5-fold. Thus, selection of suitable cultivars with low Asn contents is considered as a feasible way to minimize AA formation potential, although it has to be taken into account that environmental conditions (site-specific soil properties and climate) may alter Asn contents considerably [8,18].

Fertilization is a key measure in crop production that increases yield and quality and affects Asn levels as well. Nitrogen amount and timing of application, as well as nitrogen form can affect Asn contents in wheat considerably [21,22]. Especially high nitrogen availability during grain filling leading to high crude protein contents is considered to increase free Asn levels significantly [22]. Postles et al. [23] found for rye that free Asn was influenced by variety and nitrogen supply. Moreover, sulphur deficiency can dramatically increase Asn contents and thus the AA formation potential [24,25]. Furthermore, fungicide application promoting leaf area duration and delaying senescence can reduce free Asn content in grains [21]. Further, environmental conditions given by different growing locations can highly change the level of free Asn within wheat genotypes [19].

However, most of the studies were accomplished for conventional crop production only. Thus, the question arose if there might be a considerable difference between organic and conventional cereal production systems in their impact on AA precursor content and AA formation potential as organic farming uses different agronomic practices, like nitrogen fertilization strategies, and to some extent different cultivars. In a study by Springer et al. [26], the comparison of Asn contents of two organically and two conventionally produced rye cultivars revealed higher Asn contents in the organically produced rye but no additional information about site, growing conditions and management were given. Preliminary studies of Stockmann et al. [27] reported an effect of the cropping system, as wheat cultivars grown under organic conditions showed a significantly lower amount of free Asn when compared to conventionally grown wheat cultivars. Further studies investigating the

effect of organic versus conventional growing practice on yield, quality, and the latter AA content of cereal products have, to date, not been published.

With regard to the steadily rising demand for organically produced products in Germany and the association that organically produced foodstuffs are considered per se healthier than conventionally produced ones, filling this knowledge gap is essential.

Therefore, the main focus of this study was to compare different wheat, rye and spelt cultivars grown under conventional and organic farming conditions over two consecutive growing seasons regarding their free Asn content and AA formation potential under the consideration of yield and quality. Special emphasis was given towards analysing the impact of the management system on free Asn content and AA formation.

## 2. Materials and Methods

### 2.1. Experimental Sites

To transfer results into practice and obtain realistic results under typical management conditions of each production system, field trials were installed at two different sites located ~20 km away from each other, both in South-West Germany. The sites are long-term managed by either organic or conventional farming procedures e.g., fertilization, soil preparation and crop rotation. This must be taken into account in relation to the obtained results. The field trial under conventional management was carried out over two consecutive growing seasons (2006–2007; 2007–2008) at the experimental station Ihinger Hof, Renningen (48°44′ N 8°55′ E; average annual temperature 8.3 °C; average annual rainfall 693 mm) of the University of Hohenheim. The same trial was accomplished in parallel under organic management at the experimental station for organic farming of the University Hohenheim, Kleinhohenheim, Stuttgart (48°44′ N 9°12′ E; average annual temperature 8.8 °C; average annual rainfall 700 mm). Detailed data on temperature and rainfall during the seasons 2006–2007 and 2007–2008 for both sites are depicted in Table 1.

**Table 1.** Temperature and rainfall during the experimental seasons 2006–2007 and 2007–2008 at the locations Ihinger Hof, Renningen and Kleinhohenheim, Stuttgart, both in South-West Germany).

| Location | Ihinger Hof (Conventional) | | Kleinhohenheim (Organic) | |
|---|---|---|---|---|
| **2006/2007** | **Temp. (°C)** | **Rainfall (mm)** | **Temp. (°C)** | **Rainfall (mm)** |
| September | 16.3 | 50 | 17.4 | 59.2 |
| October | 11.9 | 74 | 12.9 | 80.0 |
| November | 7.1 | 22 | 7.8 | 25.2 |
| December | 3.4 | 22 | 4.0 | 31.2 |
| January | 4.4 | 49 | 5.0 | 22.2 |
| February | 4.4 | 69 | 5.2 | 102.0 |
| March | 5.3 | 50 | 6.5 | 80.2 |
| April | 12.2 | 1 | 13.7 | 0.2 |
| May | 14.1 | 104 | 15.0 | 142.0 |
| June | 17.7 | 107 | 17.7 | 140.8 |
| July | 16.9 | 69 | 17.8 | 91.0 |
| August | 16.4 | 64 | 17.3 | 69.0 |
| Mean/Sum | 11.4 | 681 | 11.7 | 843.0 |

**Table 1.** *Cont.*

| Location | Ihinger Hof (Conventional) | | Kleinhohenheim (Organic) | |
|---|---|---|---|---|
| **2007/2008** | **Temp. (°C)** | **Rainfall (mm)** | **Temp. (°C)** | **Rainfall (mm)** |
| September | 11.8 | 48 | 12.9 | 56.2 |
| October | 8.2 | 7 | 9.3 | 15.4 |
| November | 3.2 | 63 | 3.6 | 100.2 |
| December | 0.8 | 53 | 1.3 | 61.6 |
| January | 3.1 | 39 | 4.0 | 47.2 |
| February | 3.6 | 21 | 4.8 | 27.4 |
| March | 4.3 | 64 | 5.1 | 86.4 |
| April | 7.5 | 103 | 8.4 | 105.2 |
| May | 14.8 | 101 | 15.7 | 76.8 |
| June | 17.2 | 93 | 17.6 | 109.6 |
| July | 17.7 | 57 | 18.5 | 61.2 |
| August | 16.8 | 99 | 17.7 | 112.6 |
| Mean/Sum | 9.1 | 748 | 9.9 | 860.0 |

Soil Conditions at the Experimental Sites

The field trials at Ihinger Hof (conventionally managed) were carried out on loess derived soils with soil textures of silt (2006–2007) and silty clay (2007–2008) with sugar beet as the previous crop for both years. $N_{min}$ values at start of the vegetation period (end of March) in 2006 and 2007 in a soil depth of 0–90 cm were 2.4 and 31 kg $NO_3$-N ha$^{-1}$, respectively. The soils at Kleinhohenheim (organically managed) were loess derived with a loamy soil texture with faba beans as the previous crop for both years. $N_{min}$ values at start of vegetation period in 2006 and 2007 in a soil depth of 0–90 cm were 41.8 and 58.7 kg $NO_3$-N ha$^{-1}$, respectively.

In Table 2 the main characteristics of the soil chemical analysis for both sites are presented.

**Table 2.** Main characteristics of the soil chemical analysis for both sites over both growing seasons.

| Site | Management System | PH Value | $P_2O_5$ [mg 100 g$^{-1}$] | $K_2O$ [mg 100 g$^{-1}$] | Mg [mg 100 g$^{-1}$] |
|---|---|---|---|---|---|
| Ihinger Hof | conventional | 7.0 | 34.0 | 31.0 | 27.0 |
| Kleinhohenheim | organic | 5.8 | 8.6 | 27.0 | 13.0 |

*2.2. Experimental Design*

Ten winter wheat, five winter spelt and five winter rye cultivars (Table 3), suitable for conventional as well as for organic farming in Germany were tested in a randomized complete block design (plot size 4 × 6 m) with three replicates. To avoid neighbouring effects between the different crop species due to differences in plant height, species were separated by a border plot with a width of 2 m. Species groups were randomly placed in each block and within each species group cultivars were arranged randomly.

**Table 3.** Cereal species and cultivars tested in the field trials.

| Species | Cultivar | Quality Grade * |
|---|---|---|
| Winter wheat | Akteur | E |
|  | Bussard | E |
|  | Achat | E |
|  | Privileg | E |
|  | Capo | E |
|  | Enorm (only at Ihinger Hof) | E |
|  | Batis | A |
|  | Naturastar | A |
|  | Ludwig | A |
|  | Astron | A |
|  | Magnus | A |
| Winter spelt | Schwabenspelz, Ceralio, | - |
|  | Oberkulmer Rotkorn, | - |
|  | Franckenkorn, Schwabenkorn | - |
| Winter rye | Amilo, Nikita, Recrut | - |
|  | Danko, Pollino | - |

\* refers to the German quality classes. E-wheat: highest baking quality, A-wheat: high baking quality.

### 2.3. Agronomic Practices

Primary tillage was done with a cultivator (15 cm depth) at Ihinger Hof and with a mouldbord plough (25 cm depth) at Kleinhohenheim. Seed bed preparation was accomplished by a power harrow at both experimental sites.

Sowing was done for all species on 12 October 2006 and 09 October 2007 at Ihinger Hof and on 19 October 2006 and 17 October 2007 at Kleinhohenheim. Winter wheat and winter spelt were sown on both locations with a sowing density of 350 seeds $m^{-2}$. Sowing density of winter rye was 300 seeds $m^{-2}$. Row distance was 0.10 m at Ihinger Hof and 0.12 m at Kleinhohenheim.

Nitrogen was applied as CAN (calcium ammonium nitrate: 13.5% Nitrate-N, 13.5% Ammonium-N) in the conventional trial. For winter wheat, total nitrogen amounts of 190 to 195 kg N $ha^{-1}$ were applied in four to five rates according to the expected yield and the expected crude protein content (for details see Table 4). Winter spelt and winter rye cultivars were fertilized with 140 and 150 kg N $ha^{-1}$ in four rates (Table 4). No further nutrients were applied during this field trial. However, during a 3–4 yr crop rotation a general fertilization of sulphur, phosphorous and potassium was carried out to maintain soil fertility.

**Table 4.** Time and amount of the N-fertilization for the conventionally produced species. The two dates refer to the application date for the single growing season.

| Species/Quality Class | Time and Amount of N-Fertilization [kg N $ha^{-1}$] | | | | | |
|---|---|---|---|---|---|---|
|  | $N_{Total}$ | $N_{Start\ of\ Veg.}$ | $N_{EC31/32}$ * | $N_{EC37/39}$ | $N_{EC49/51}$ | $N_{EC59/61}$ |
|  |  | 08/03/2007 | 02/04/2007 | 07/05/2007 | 24/05/2007 | 31/05/2007 |
|  |  | 06/03/2008 | 28/04/2008 | 28/05/2008 | 02/06/2008 | 10/06/2008 |
| E-Wheat | **190** | 40 | 40 | 30 | 50 | 30 |
| A-Wheat | **195** | 55 | 40 | 50 | 50 | - |
| Spelt | **140** | 40 | 40 | 30 | 30 | - |
| Rye | **150** | 70 | 40 | - | 40 | - |

\* EC: Growing stage of the plant.

In the organic trial, all species were supplied with 100 kg N $ha^{-1}$ by liquid cattle manure (100 $m^3$ $ha^{-1}$: 1 kg N $m^{-3}$, total nitrogen content, 4% dry matter) in two rates a 50 $m^3$ $ha^{-1}$ at start of vegetation and at start of stem elongation.

Growth regulators (Chlorcholinchloride and Trinexapac-ethyle), herbicides (Atlantis®: Iodsulfuron-methyl-sodium, Mefenpyr-diethyl, Mesosulfuron-methyl, Concert®: Metsulfuron-methyl, Thifensulfuron-methyl, Primus: Florasulam), fungicides (Juwel top®: Fenpropimorph, Epoxiconazol, Kresoxim-methyl) and insecticides (Bulldock®: beta-Cyfluthrin) were broadcast as needed in the conventional trial. No pesticides and no growth regulators were applied in the organic trial.

In 2008, the organic field was harrowed twice (31 March 2008 and 26 May 2008) to control weeds. Due to low weed densities, harrowing was omitted in the year prior.

Harvest was accomplished by a Hege 180 plot combine harvester (Hege, Eging am See, Germany) after grains had reached a dry matter content of 85%.

### 2.4. Yield

Grain yield of the cereal species and cultivars was determined by weighing the plot yield. Grain samples were dried at 105 °C for 24 h to determine grain moisture. Grain yields given refer to 86% dry matter content.

### 2.5. Flours

For the determination of quality parameters, the determination of the AA precursor content free Asn and the AA formation potential, grain samples were milled on a laboratory mill (Quadrumat Junior, Brabender, Duisburg, Germany) to obtain white flours. Ash content of flours was about 0.5% of flour DM. Flour moisture was calculated from the weight loss before and after drying of about 5 g flour at 105 °C for 24 h.

### 2.6. Quality Analysis

#### 2.6.1. Crude Protein Content

Total grain nitrogen content was determined by Near-Infrared-Spectroscopy (NIRS, NIRS 5000, FOSS GmbH Rellingen, Germany). Calibration samples were analysed according to the Dumas Method [28] using a Vario Max CNS analyzer (Elementar, Hanau, Germany). The analysed final nitrogen content was multiplied by a factor of 5.7 [29] for wheat samples and 6.25 for spelt and rye samples.

#### 2.6.2. Zeleny's Sedimentation Test

Zeleny´s sedimentation test was determined in wheat and spelt flours using 3.2 g flour according to ICC standard No. 116. The sedimentation values of the flours were adjusted to a 14% moisture basis.

#### 2.6.3. Free Asparagin

Free amino acids were extracted with either 2 g of wheat or spelt flour or 1 g of rye flour and were mixed with 8 mL of 45% ethanol for 30 min at room temperature. After centrifugation for 10 min at room temperature with 4000 rpm and 10 min at 10 °C and 14000 rpm, the supernatant was filtered through a 0.2 μm syringe filter and poured into vials. Asparagine analysis was performed using Merck-Hitachi HPLC components. The pre-column derivatization with FMOC [30] was completely automated by means of an injector program. Subsequently, the derivatized Asn was separated on a LiChroCART Superspher RP 8 column (250 mm × 4 mm, Fa. Merck, Darmstadt, Germany) at a constant temperature of 45 °C. The fluorescence intensity of the effluent was measured at the excitation and emission maxima of 263 and 313 nm.

#### 2.6.4. Acrylamide Formation Potential

The AA formation potential of cereal flour was assessed according to the AA contents of 5 g flour in 250 mL Erlenmeyer flasks after heating in an oven for 10 min at 200 °C. Due to the complexity of the

acrylamide analysis, sample size was reduced to an overall number of 28 samples (6 winter wheat, 4 winter spelt and 4 winter rye samples from two investigation years).

Sample preparation was accomplished according to the test procedure 200L05401 [31] of the Chemische und Veterinäruntersuchungsamt (CVUA) Stuttgart.

After cooling down to ambient temperature, 100 mL of bidestilled water and 100 μL of $D_3$-Acrylamide were added as an internal standard to the heated flour samples in the Erlenmeyer flasks. To completely extract acrylamide from the flour, samples were put in an ultrasonic bath for 10 minutes at 40 °C. After adding 1 mL of Carrez I and II to each of the samples and shaking the flasks thoroughly, the samples were filtered using folded filter paper to separate the colloids and flour particles from the aqueous solution. Subsequently, samples were cleaned up by a solid phase extraction in a vacuum chamber after preconditioning the cartridges by 10 mL of bidestilled water and 10 mL methanol. After sample clean-up, about 1 to 2 mL of the eluate from each sample was filled in an autosampler vial and was deep frozen (−18 °C) until AA was determined by LC-MS-MS by the CVUA according to the test procedure 201L01301 [32]. The eluates were separated by a graphite or RP18-phase and detected by tandem-mass-spectrometer. Quantification was undertaken by using the isotope-labelled internal standard ($D_3$-Acrylamide).

### 2.7. Statistical Analysis

Yield and quality data (crude protein, sedimentation value), as well as free Asn were subjected to analysis of variance (ANOVA) using the Procedure MIXED from the statistical software package SAS 9.1. (SAS Institute Inc., Cary, NC, USA). If necessary, data were ln- or square root-transformed, to stabilize normal distribution and homogeneity of variance. A comparison of means was accomplished using the *t*-Test.

ANOVA was done in two steps: In a first step, the main effects year, management system, crop species and interactions were investigated. In a second step, crop species were analysed separately for determining potential varietal differences depending on year and management system. Only for the parameter AA, no statistical analysis was undertaken. This was because only single samples from one field replicate were selected by the level of free Asn to represent the whole range from low to high for each species. The analyses of AA were chosen to reveal if free Asn can serve as indicator for AA formation potential as it was reported by several studies.

## 3. Results

Grain yield, quality parameters and free Asn content of the three cereal species were significantly influenced by management system and year (Table 5). Additionally, the threefold interaction $CS \times Y \times S$ was significant for all parameters except for the sedimentation value.

**Table 5.** F-values and *p*-values for main effects: management system (MS), species, year and interactions between factors on grain yield, crude protein content, sedimentation value and the Asn content of the flours.

| | DF [1] | Grain Yield | | Crude Protein | | Sedimentation Value | | Free Asn | |
|---|---|---|---|---|---|---|---|---|---|
| | | F | *p* Value * | F | *p* Value | F | *p* Value | F | *p* Value |
| **MS** | 1 | 101.92 | *** | 266.94 | *** | 15.41 | *** | 41.60 | *** |
| **Species (S)** | 2 | 40.77 | *** | 710.29 | *** | 118.62 | *** | 470.25 | *** |
| **Year (Y)** | 1 | 1.78 | 0.21 | 60.33 | *** | 0.65 | 0.42 | 52.01 | *** |
| **MS × S** | 2 | 1.79 | 0.17 | 69.49 | *** | 19.59 | *** | 7.29 | *** |
| **MS × Y** | 1 | 1.47 | 0.25 | 73.02 | *** | 0.05 | 0.82 | 0.02 | 0.89 |
| **S × Y** | 2 | 11.44 | *** | 1.83 | 0.16 | 4.46 | * | 10.45 | *** |
| **MS × S × Y** | 2 | 8.53 | *** | 32.66 | *** | 0.13 | 0.71 | 14.89 | *** |

* level of confidence ($p < 0.05$ *, 0.01 ** or 0.001 ***), [1] degree of freedom.

Winter wheat (WW) and winter spelt (WS) grain yields were about 1.1 to 1.9 t ha$^{-1}$ higher in the conventional system when compared to the organic management systems in both experimental years (Table 6). In contrast, conventionally produced winter rye only achieved higher yields when compared to the organic management system (by 2 t ha$^{-1}$) in 2008.

For both years, crude protein content of winter wheat was significantly higher under conventional conditions (Table 6). The mean protein content of conventionally grown samples was 13.37% compared to 9.72% if organically produced and by this 27% higher. Additionally, a significant year-to-year effect was obvious. In contrast, crude protein content of spelt was not significantly affected by the production system. Crude protein contents of conventionally grown rye were only significantly higher in 2007. In general, the year had a higher impact on the conventionally managed cereals. In 2007, the protein level was around 10% higher.

Sedimentation values of winter wheat for both years were considerably higher under conventional growing conditions when compared to organic farming (Table 6). For spelt, no effect was observed for either years.

**Table 6.** Grain Yield (GY), quality parameters (crude protein [CP], sedimentation value [SV]) and free Asn) of the three tested species depending on management system and year. Treatments within the same species (WW = winter wheat, WS = winter spelt, WR = winter rye) assigned by the same letters are not significantly different ($\alpha$ = 0.05, *t*-Test).

| | | GY | | CP | | SV | | Asn | |
|---|---|---|---|---|---|---|---|---|---|
| | | [t ha$^{-1}$] | | [%] | | [mL] | | [mg 100 g$^{-1}$ DM] | |
| | | Conventional | Organic | Conventional | Organic | Conventional | Organic | Conventional | Organic |
| **WW** | **2007** | 7.70 c | 6.55 b | 14.18 c | 9.77 a | 37.00 b | 29.09 a | 16.07 c | 12.67 b |
| | **2008** | 7.01 b | 6.04 a | 12.56 b | 9.66 a | 40.47 b | 31.56 a | 14.97 bc | 7.89 a |
| **WS** | **2007** | 6.44 c | 4.56 a | 14.78 b | 14.46 b | 25.40 a | 25.33 a | 10.47 ab | 12.24 b |
| | **2008** | 6.75 c | 5.50 b | 13.16 a | 13.23 a | 24.03 a | 24.81 a | 11.44 b | 8.62 a |
| **WR** | **2007** | 7.54 bc | 7.22 b | 10.74 c | 6.47 a | | | 54.31 c | 35.92 b |
| | **2008** | 7.87 c | 5.83 a | 7.54 b | 7.35 b | | | 31.50 ab | 30.14 a |

### 3.1. Free Asn as Main AA Precursor

Free Asn content of the analysed white flours was significantly affected by management system, species, year and the interaction between these effects (Table 5). Rye had the highest Asn contents in both management systems and for both years with on average about 38 mg 100 g$^{-1}$ flour-DM, followed by winter wheat with 13 and winter spelt with 11 mg 100 g$^{-1}$ flour-DM (Table 6). Averaged across the three species, organically produced grain contained about 26% lower Asn contents compared to conventional production. Differences between management systems were most distinctive in winter wheat, with 22% to 47% lower Asn contents under organic production depending on the year (Table 6). Summing up years level of free Asn for conventional wheat samples were 15.52 mg 100 g$^{-1}$ flour-DM and for organic wheat samples were 10.28 mg 100 g$^{-1}$ flour-DM. Thus, the organically cropped wheat samples had a 33% lower level of free Asn. Asn contents of spelt and rye differed only significantly between the two management systems in single years (Table 6, Figure 1). Whilst Asn levels of spelt were not affected by the management system in 2007, organically produced spelt had a significantly lower Asn content of 8.6 mg 100 mg$^{-1}$ flour-DM in 2008, which was about 25% lower than under conventional farming for the same year. However, spelt cultivars revealed to have significant effects on Asn levels irrespective of year and management system, with cultivars Frankenkorn and Schwabenkorn having the lowest and highest Asn contents (Franckenkorn: 7.5 mg 100 mg$^{-1}$ flour-DM, Schwabenkorn: 13 mg 100 mg$^{-1}$ flour-DM), respectively (Figure 1). In 2007, organically produced rye had significantly lower Asn contents while no significant differences in the Asn content occurred between the two management systems in 2008 (Table 6). By comparing the Asn response of the different cultivars in both management systems, four out of ten organically produced wheat cultivars had significantly lower Asn levels than their conventional counterparts. In 2007 and in 2008, Asn contents of each of the ten tested wheat cultivars were significantly lower under organic compared to conventional farming (Figure 1). In this context an interesting observation was the different responses of each cultivar to the management system concerning free Asn. Whereas free Asn of cv. Bussard hardly changed between both management systems in both years and in 2008 was even slightly higher in the organic trial, cv Privileg changed a lot. In between, cv. Naturastar showed small difference of free Asn in 2007 but a large difference in 2008.

In 2007, cvs Privileg and Naturastar had the highest Asn contents of 26.6 and 24.5 mg 100 g$^{-1}$ under conventional conditions. All other conventionally grown cultivars had significantly lower levels than Privileg and Naturastar in a range between 12 and 16 mg 100 g$^{-1}$ but did not differ significantly from each other (Figure 1). Cultivars Privileg and Naturastar also showed the highest Asn contents under organic conditions (18.5 and 24.8 mg 100 g$^{-1}$), which were significantly higher than from organically produced cultivars Bussard, Achat, Batis, Astron and Magnus with Asn contents ranging between 11.5 and 14 mg 100 g$^{-1}$ flour-DM. In 2007, the significantly lowest Asn value under organic farming was recorded for cultivar Ludwig with 6 mg 100g$^{-1}$ followed by cultivar Capo with 9.1 and cultivar Akteur with 9.8 mg 100 g$^{-1}$ flour-DM (Figure 1). These three were the only cultivars with Asn contents lower than 10 mg 100 g$^{-1}$ flour-DM, whereas under conventional farming none of the tested cultivars had Asn contents below this level. Privileg and Naturastar were the cultivars with the highest Asn contents, as well as in 2008, under conventional farming. With levels of 31.6 and 32.1, they significantly outreached the levels of all other conventionally produced wheat cultivars, which were in a range between 10.9 and 14 mg 100 g$^{-1}$ flour-DM, except cultivar Capo which had the significantly lowest Asn content of about 10.1 mg 100 g$^{-1}$ flour-DM (Figure 1). Asn levels of all organically grown cultivars were lower than 10 mg 100 g$^{-1}$ flour-DM, except for cultivar Privileg with 10.7 mg 100 g$^{-1}$ flour-DM. Cultivars Capo, Astron and Achat had the significantly lowest Asn contents of all organically produced wheat cultivars in 2008 with values of 6.1 and 6.5 mg 100 g$^{-1}$ flour-DM.

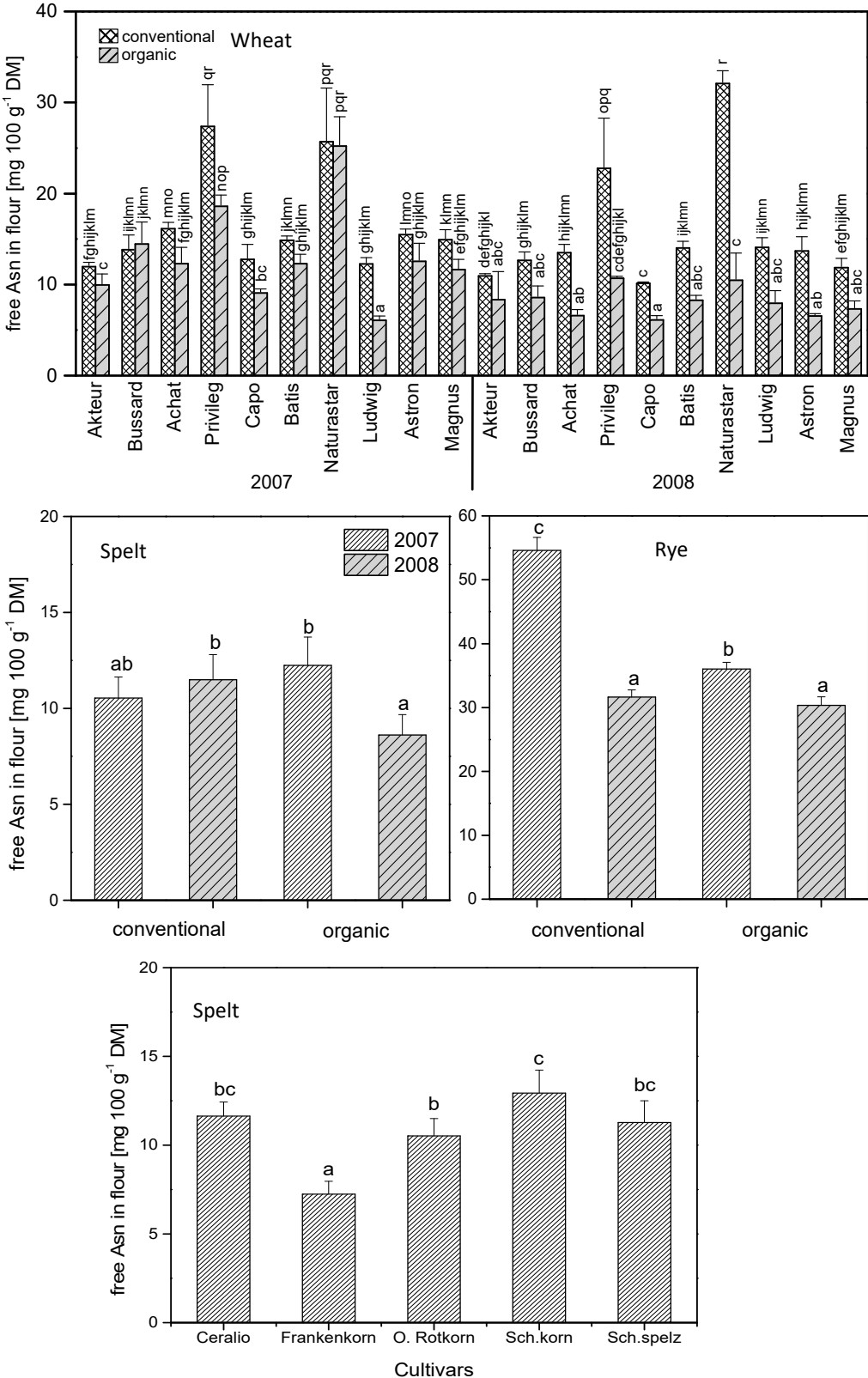

**Figure 1.** Free Asn content separated by species (wheat, spelt, rye), cropping system (organic, conventional), year (2007, 2008) and cultivars in dependence of their significant highest order interaction. Columns with different letters within cropping system and year indicate significant differences ($\alpha = 0.05$, *t*-Test).

### 3.2. Correlation of Crude Protein, Free Asn and AA

Between flours derived from the different wheat cultivars there was only a weak correlation of both parameters between different cultivars under both management practices in both years ($R^2 = 0.18$, Figure 2), meaning that cultivars with high crude protein contents did not show high Asn levels per se and vice versa. The correlation of crude protein and Asn of winter spelt was weak across the two management systems and years ($R^2 = 0.1$, data not shown) when averaged over cultivars, which was mainly due to a low variability of crude protein content between the management systems and years. Crude protein content and Asn content of rye correlated well with a $R^2 = 0.8$ (data not shown).

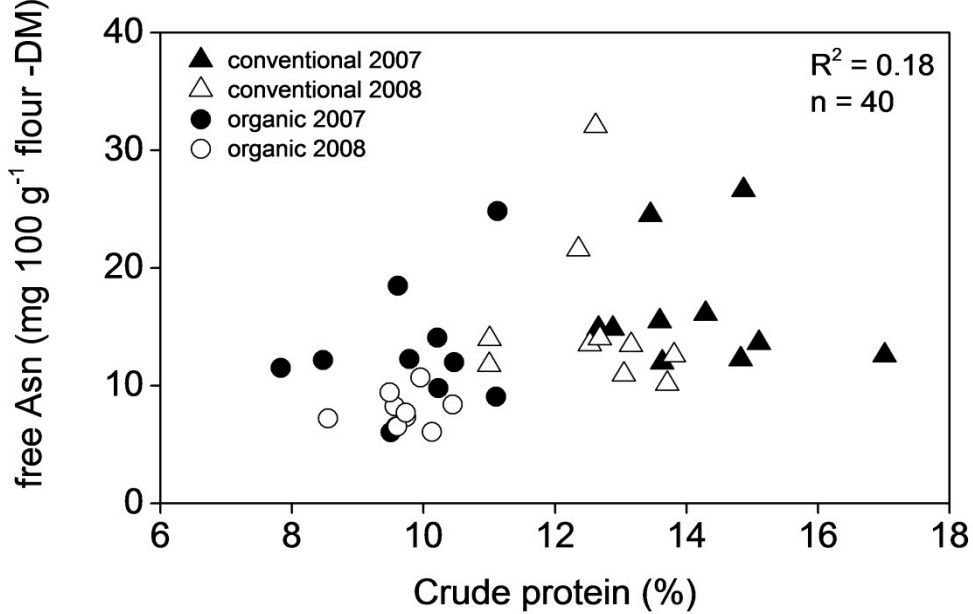

**Figure 2.** Relationship between crude protein content and free Asn levels in wheat flours produced under conventional and organic conditions in 2007 and 2008.

Free Asn and AA across management systems, years and species correlated well by $R^2 = 0.64$ (Figure 3A). Separating the management systems led within both systems to a strong relation of both traits (Figure 3B). The conventional crop management system showed an $R^2$ of 0.58, while for the organic management system the relation was even stronger with $R^2 = 0.75$. For wheat, both traits correlated with $R^2 = 0.82$ in 2007 and $R^2 = 0.5$ in 2008 for the conventionally produced samples and with $R^2 = 0.76$ in 2007 and $R^2 = 0.86$ in 2008 for the organically produced wheat flours.

Across management systems, free Asn and AA of spelt correlated by $R^2 = 0.55$. For organic samples, only the relationship was strong by $R^2 = 0.83$ but low for the conventional ones ($R^2 = 0.4$). The correlation within rye samples was generally lower by $R^2 = 0.37$ (across management systems), $R^2 = 0.25$ (organic samples) and $R^2 = 0.5$ (conventional samples).

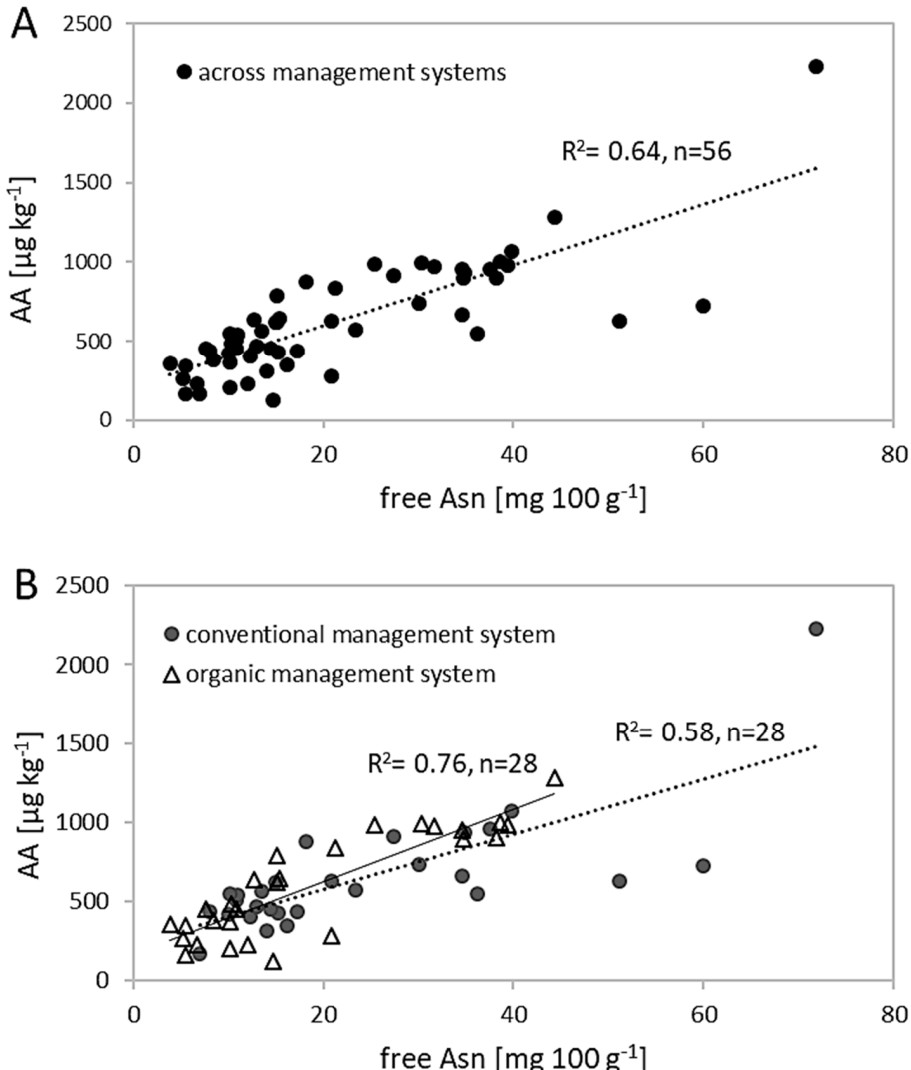

**Figure 3.** Relationship between free Asn in wheat flours and AA in heated wheat flours across management systems (**A**) and separated by conventional vs. organic management system (**B**).

## 4. Discussion

### 4.1. Grain Yield and Baking Quality

Grain yield of all three organically produced species was significantly lower compared to their conventionally grown counterparts. Organically grown wheat produced on average about 85% of the yield compared to conventionally gown. For spelt and rye the yield gap was even larger. Differences concerning grain yield of 20% to 30% between conventional and organic farming are described by other authors [33–35]. Determined lower protein contents of on average 3.5% across both years between conventionally and organically produced wheat corresponded well with results reported by Bilsborrow et al. [36] from a long-term field trial in the UK. Both difference in yield and crude protein content were most likely the result of differences in the availability of plant-available nitrogen between the two management systems as shown by other authors [36,37]. Especially, nitrogen availability influences grain yield [38] and crude protein content [22], which is an important quality parameter for baking quality of wheat due to its strong correlation with bread volume. Nitrogen fertilizer amounts in conventional farming are usually higher, mineral nitrogen is available faster than organic nitrogen. Thus, mineral nitrogen in conventional farming can be directed more precisely, leading to higher grain yields. Under organic conditions, there is often a mismatch between the plant-nitrogen demand and

the mineralisation rate depending on environmental conditions e.g., temperature and soil moisture [39]. Moreover, higher weed, pest and disease pressure could also have been responsible for yield differences but were not assessed in our experiment as measures to control them under organic farming are very limited. Moreover, site-specific conditions usually strongly affect yield more than weeds, pests and diseases when crops are grown organically [40]. Thus, the manifold regulatory options in conventional management systems can compensate for location deficits better, leading to higher grain yields [41]. Consequently, if organically produced bakery goods are demanded, lower yields and lower baking qualities must be accepted [42], and bread bakery processing has to be adjusted to the lower protein contents of such flours [43] to achieve acceptable products.

However, the postulated levels of protein needed by industry are under discussion as there are some studies announcing that good baking quality of, for example, wheat flour, does not need high levels of crude protein [44]. Much more important are gluten content, gluten quality and sensory property. Thus, wheat cultivar cultivated under organic farming conditions can offer a good baking quality while having lower protein content. Furthermore, improving baking quality means also improving sensory properties. Torri et al. [45] reported that a treatment with mycorrhizal factors is able to raise the sensory properties of modern wheat varieties. No significant difference of spelt crude protein content between the two management systems could be observed. This was presumably due to only minor differences in nitrogen fertilizer amounts and nitrogen availability especially in the latter phase of grain filling.

*4.2. Free Asn*

Free Asn is the critical factor for acrylamide formation during processing of cereal based bakery products [12], and Asn contents in flour correlate almost linearly with the acrylamide content in heated flour [8] or in breads [7]. Free asparagine also proved to be strongly correlated with the AA formation in heated flour samples in our study. Rye Asn contents were considerably higher than that of wheat and spelt. This corresponds well with findings from Fredriksson et al. [46], Elmore et al. [47] and Claus et al. [7] who found up to 3 or 4 times higher Asn levels in rye compared to wheat or spelt. Therefore, the acrylamide formation potential of bakery goods made from rye flour is considered to be higher than from wheat or spelt [7,47]. In contrast studies by Curtis et al. [8] showed that AA formation in heated flour per unit Asn was much lower in rye than in wheat flour, suggesting that the higher Asn level compared to wheat does not inevitably mean that rye has a higher AA formation potential per se.

The amount of free Asn was strongly affected by the management system. Across all species, free Asn was 26% lower under organic conditions. Differences in the Asn level between the management systems were consistent across both experimental years for wheat, whereas spelt and rye showed inconsistencies. Organically produced wheat had a lower Asn content of about 33% compared to conventional production when averaged over years and cultivars, whereas Asn contents of organically produced spelt and rye were lower only in single years. Therefore, organically produced wheat particularly offers the opportunity to significantly lower the AA potential of bread and bread rolls via choosing a raw material low in free Asn.

However, in particular for wheat also, cultivars had a significant impact on free Asn formation. While cv Bussard showed only a slight difference between management systems in both years, cv Privileg changed to a large content in 2007 and 2008. Interestingly, cv Naturastar seemed to be highly influenced by year, as in 2007 the differences between management systems were hardly given while in 2008 the distinction was huge. The reason that the cvs reacted in three different ways could be the quality class of the cvs. Referring to the German quality classes Bussard and Privileg belong to E-wheats (highest baking quality) while Naturastar belongs to A-wheats (high baking quality). Thus, E- and A-wheats differ in the protein synthesis capacity. Furthermore, climate conditions (temperature and rainfall) during grain filling (June–August) may have influenced free Asn level in those cvs. However, only small changes in both, temperature and rainfall from June to August were observed in this study. Nevertheless, several studies showed that climate conditions (growing temperature,

sunshine duration and water availability) during grain growth can highly influence Asn formation, protein synthesis, amino acid composition, nitrogen remobilization and phytochemicals in cereal grains [7,8,18,48,49]. In this context, further research especially revealing the reason why cultivars differ in their Asn formation potential is essential.

Weber et al. [22] investigated the impact of different nitrogen amounts and timings on the resulting Asn levels in flours from a conventionally produced winter wheat cultivar. They found a close correlation between crude protein and Asn content. Furthermore, it was predominantly the 'late applied nitrogen' that led to crude protein contents of more than 13% which significantly increased free Asn compared to the unfertilized control [22]. Winkler and Schön [50] found a close correlation between nitrogen concentration and free Asn in barley grain, as well as Curtis et al. [8] found in rye grain. As nitrogen fertilization in conventional farming is usually applied at higher amounts, and availability of mineral nitrogen is also higher than found in organically applied nitrogen, the higher level of free Asn under conventional farming presumably resulted from the differences in the nitrogen supply and availability between both systems during the phase of grain filling. In this context, it is quite likely that applying lower nitrogen in the conventional trial would match the organic trial, with the same lowering effect of protein levels as well [51,52]. However, it seems not to be that easy as a set of organically cropped cereal species and cultivars were investigated by Stockmann et al. [53] for their content of free Asn. The samples were only marginally supplied with nitrogen, but a high range of free Asn comparing species and cultivars within species was observed. Moreover, a broad range of crude protein was reported (8.7% to 13.7% across years and wheat cultivars). Thus, reducing nitrogen to reduce levels of free Asn appears to be a feasible way, but some other factors must be considered too. Finally, we suggest that the level of nitrogen available for plants must fit to the synthesised crude protein level to lower free Asn.

In general, winter spelt and winter rye have a lower N-demand compared to wheat. This led to a smaller difference in the crude protein contents within the two species between the conventional and organically grown grains. Thus, this may explain the less pronounced impact of the management system on the Asn contents within winter spelt and winter rye in a single year. Springer et al. [26] found higher amounts of Asn in organically produced cultivars compared to those conventionally produced. They assumed that the higher share of the seed coat on the grain under organic cultivation might have been responsible for the higher contents of Asn found in the organically cropped cultivars. This effect could not be observed in our study. Nevertheless, as reported by Corol et al [19], wholemeal flour, can reach higher levels of free Asn in a range of up to 32 to 156 mg 100 $g^{-1}$ which is about 6 to 10-fold above our reported results within wheat flour.

Differences in the free Asn content between management systems especially in wheat could also be caused by diseases caused by fungus. Martinek [21] found that fungicide application promoting leaf area duration and delaying senescence can reduce free Asn content in grains. In our study, only in the conventional trial fungicides were applied. Thus, the level of free Asn could have been raised in the organic samples. However, as the level of free Asn in general was lower in organic samples, impact of nitrogen seems much higher than the effect of fungicides. However, the infestation of plant leaf by fungus should be considered as causing higher levels of free Asn. Cultivars owning a high baking quality do not inevitably have to be linked with high Asn contents and ultimately a high AA formation potential. This is suggested by the weak correlation found between crude protein and free Asn contents of different wheat cultivars. Furthermore, the results showed that cultivars combining high or acceptable baking quality and low Asn contents are already available and that the AA formation potential could further be lowered by selecting and cultivating appropriate cultivars, irrespective of management systems, if the information on genotypic disposition regarding Asn accumulation would be available.

Sulphur can have a high impact on free Asn in case of a deficiency in soil [25]. Within the conventional system, only mineral nitrogen fertilizer CAN was applied but no additional sulphur. In contrast, the organic management system was fertilized by manure that can contain sulphur. Thus,

some differences concerning free Asn between both management systems could have been caused by the influence of sulphur. However, this was not analysed in detail.

Moreover, it has to be considered that environment (=location and year) can affect Asn levels considerably, as also shown by Curtis et al. [18] for wheat and by Curtis et al. [8] for rye. However, up to now information on how soil type, temperature and precipitation affect grain-Asn accumulation is missing [54]. Within wheat genotypes grown at six locations in Europe, Corol et al. [19] described that only 13% of the free Asn variability was explainable by heritability while about 36% was caused by environment. This must be also taken into account for the interpretation of our data, as the locations for the conventional and for the organic trials were located within a distance of 20 km. Therefore, cultivars must be tested at different locations for at least two years before recommendations or selection of cultivars can be announced for further specific breeding programs targeted towards lowering the AA precursor contents.

## 5. Conclusions

The study aimed to assess the effect of organic versus conventional farming practices on free Asn contents in flours of winter wheat, winter spelt and winter rye. The management system significantly affected Asn levels. The effect was most noticeable and consistent for winter wheat, where Asn levels were lowered on average by 33% if cultivars were organically grown. Organically grown spelt and rye showed lower Asn contents only in single years. Different intensities of plant-available nitrogen during critical phases of grain development between both systems are considered to be mainly responsible for differences in Asn accumulation. These differences were most evident for wheat due to its high N demand and the close correlation between nitrogen supply and baking quality. Therefore, organically produced wheat flour can be well regarded as having a lower AA formation potential than conventionally produced wheat, despite also having lower yields. Furthermore, the weak correlation between crude protein and Asn content between different cultivars suggested that choice and cultivation of cultivars combining low Asn contents with adequate baking quality can help in lowering the amount of dietary AA intake from cereal based bakery goods, irrespective of whether the raw material originates from organic or conventional farming. Especially low-Asn cultivars with high stability over different environments are of great interest for future research in this area.

**Author Contributions:** Conceptualization, F.S., E.A.W., W.C. and S.G.-H.; methodology, F.S. and E.A.W.; formal analysis, F.S., B.M.; investigation, F.S., B.M., N.M. and P.S.; resources, W.C.; data curation, F.S. and E.A.W.; writing—original draft preparation, F.S.; writing—review and editing, E.A.W. and S.G.-H.; visualization, F.S.; supervision, S.G.-H. and W.C.; funding acquisition, F.S.

**Funding:** This research was funded by a scholarship to Falko Stockmann by the Faculty of Agricultural Sciences of the University of Hohenheim.

**Acknowledgments:** The authors would like to thank the technical staff of the experimental station "Ihinger Hof" and "Kleinhohenheim" for the agronomic management of the field trials.

**Conflicts of Interest:** The authors declare no conflict of interest. The funders had no role in the design of the study; in the collection, analyses, or interpretation of data; in the writing of the manuscript; or in the decision to publish the results.

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
