# Peer review of "Acrylamide-Formation Potential of Cereals: What Role Does the Agronomic Management System Play?"

_agronomy, doi:10.3390/agronomy9100584_

Round 1

Reviewer 1 Report

This is an interesting study that adds to our knowledge of the importance of crop management to the asparagine content of cereal grains and acrylamide formation in cereal products. I have a few suggestions:

I think the title could be improved. Perhaps 'Acrylamide-formation potential of cereals: what role does the cropping system play? Around line 150 where N application rates are given: what else was applied to the conventional field trial: sulphur, P, K? The S supply in particular is important because the manure applied to the organic trial would contain some S. 169 Flours I noted that asparagine levels were quite low but only at the end realised that white flour was being analysed (is that correct?), which would explain it. This needs to be made clear at the start of the results and in the abstract, since it makes such a big difference.  Asparagine concentrations should be given in mmol per kg Line 254. a 1.4% higher crude protein content, is ambiguous. I think the levels in 2007 were approx. 10% higher than in 2008.  Figure 1. An interesting observation from this figure is the different responses of each cu;ltivar to the cropping system. Bussard hardly changes at all, for example, and is actually slightly higher in the organic trial, whereas Privileg changes a lot. Naturastar shows little difference in 2007 but a large difference in 2008. This should be discussed. I don't think Figure 2 is useful. There are only 4 points. How was the 'bestfit' line calculated? Clearly very different lines would fit the data quite well. Discussion. It is quite likely that applying lower N in the conventional trial would match the organic trial, with the same consequence for protein levels as well.  Some recent references have been omitted, for example Curtis et al. (2016) J Agric Food Chem 64, p9689; Curtis et al (2018) Food Chem 239, p304.

Author Response

Dear reviewer 1,

Kind regards,

Falko Stockmann

Reviewer 2 Report

Dear authors I read with interest your paper.

This paper evaluate the impact of the cropping system (conventional vs. organic farming) on AA precursor levels of free asparagine (Asn) across different cultivars of the cereal species, namely winter wheat (Triticum aestivum), winter spelt (Triticum aestivum ssp. spelta) and winter rye (Secale cereale) with simultaneous consideration of gained grain yields and flour qualities.

The objective is interesting and within the scope of the journal.

The introduction is clear and well referenced.

The material and methods are structured although there are some limiting factors e.i. different locations for the 2 managements systems with some difference in treatments and practices. So could be that too much variability create statistical difference but effected by error.

Discussion and conclusion are quite good.

Bibliography ranking needs corrections as there is numerical NO order!

Table numbering needs correction and I would recommend to transforms some tables into graphs

Specific Comments:

Title and everywhere

I’d change the world “cropping systems” with “management systems” as the difference should be based on different management (organic vs conventional). Otherwise I found difficult to understand the whole experiment. The crops are the same.

Abstract

Row 18-20: I don’t understand the phrase. May be you are referring only to Wheat?

Intro

Row 41: how is possible that you start with reference n 10?!

M&M

Row 109: what does it means “The sites are characteristic for organic and conventional farming procedures.”?

It is clear that the different agronomic practices effect the results, but how to interpret?

111: Why you publish these results only today after 10 years?

Are they still valid?

Maybe transform Table 1 into a graph?

120-124: Soil condition must be much more details and not only about N mineral. What about total N and C total? Please add a table with main characteristics of soil chemical analyse.  

Please remember that the Ihinger Hof is the ORGANIC trials and the Kleinhohenheim is the Conventional.

129-130: How can you affirm that “Ten winter wheat, five winter spelt and five winter rye cultivars (Table 2), suitable for conventional, as well as for organic farming in Central Europe” please add a reference

135: please change Table 1 into Table 2

137: please change 2.3. Experimental performance into 2.3. Agronomic practices or treatments

138-144: why the soil treatments and machine, soil density are different? This of course will effect the trials and results!

150: please change Table 2 into Table 3. What the numbers next to N means? What the 2 dates means? Why E-Wheat receives less N that A-wheat? I’d suppose that more quality wheat will produce more proteins and so need more N. isn’t it?

169: What F.Lourse means? please change the title into “quality analysis”

175-213: I’d delite all this paragraph  

223-225: thisis of course a quite strong limiting factor

Results

230: please change Table 3 into Table 4. Explain DF as degree of freedom

334: I’d change this table into a more summarising graph

Discussion

363: I suggest to add “Torri T, Migliorini P, Masoero G (2013). Sensory test vs. electronic nose and/or image analysis of whole bread produced with old and modern wheat varieties adjuvanted by means of the mycorrhizal factor. Food Research International 54:1400–1408”

Author Response

Dear reviewer 2,

Kind regards,

Falko Stockmann
